# Scale-Dependent Dielectric Properties in BaZr_0.05_Ti_0.95_O_3_ Ceramics

**DOI:** 10.3390/ma13194386

**Published:** 2020-10-01

**Authors:** Leontin Padurariu, Vlad-Alexandru Lukacs, George Stoian, Nicoleta Lupu, Lavinia Petronela Curecheriu

**Affiliations:** 1Dielectrics, Ferroelectrics & Multiferroics Group, Faculty of Physics, “Alexandru Ioan Cuza” University of Iasi, 11 Carol I Boulevard, 700506 Iasi, Romania; leontin.padurariu@uaic.ro (L.P.); vlad.lukacs@uaic.ro (V.-A.L.); 2National Institute of Research and Development for Technical Physics, 700050 Iasi, Romania; gstoian@phys-iasi.ro (G.S.); nicole@phys-iasi.ro (N.L.)

**Keywords:** ceramics, grain size, dielectric permittivity, finite element method

## Abstract

In the present work, BaZr_0.05_Ti_0.95_O_3_ ceramics with grain sizes between 0.45 and 135 µm were prepared by solid-state reaction and classical sintering. The effect of grain size on dielectric properties was systematically explored, and it was found that dielectric permittivity reaches a maximum value for grain sizes between 1.5 and 10 µm and then rapidly drops for larger grain sizes. A numerical finite element method was employed to eliminate the effect of porosity on the effective values of permittivity. The results indicate that it is possible to have a critical size in slightly doped barium titanate ceramics with enhanced functional properties for a grain size between 1.5 and 10 µm.

## 1. Introduction

Among the wide class of smart multifunctional materials, BaTiO_3_ (BT) is widely used in microelectronics in applications as multilayer ceramics capacitors (MLCCs), sensors, transducers, wireless devices for reconfigurable microwave circuits, varactors, tunable filters, phase shifters, etc. [1,2] due to its dielectric, piezo, pyro, ferroelectric and electro-optic properties. The demand for high-volume capacitance in MLCCs, required by the tremendous need for miniaturization, has led to the necessity to investigate the properties of BT ceramics with different grain sizes. This technology-driven interest in size-dependent phenomena at nanoscale motivated a number of important studies and review articles [3,4,5], including our extensive previous works on nanostructured BT ceramics [6,7,8]. On the other hand, in recent years there has been a revival of interest concerning the critical size of about 1μm, at which enhanced functional properties are reported, as a general trend [9,10,11]. In the case of pure BT, this critical size has been attributed to internal residual stress, peculiar domain density, domain wall motion or phase superposition [12,13].

While in the case of pure BT ceramics, the size-dependent properties have been extensively studied, the effect of grain size (GS) on functional properties of BT-based compositions has only recently been reported [14,15]. Mostly, the works reported different processing techniques for obtaining dense nanostructured ceramics such as (Ba,Sr)TiO_3_ [16] or Ba(Zr,Ti)O_3_ [17], analyzing a narrow grain size range and large dopant addition. Recently, Dupuy et al. reported the effects of grain size and phase homogeneity on the ferroelectric properties of a more complex solid solution of BT—(Ba,Ca)(Ti,Zr)O_3_ [18]. However, the scale-dependent properties in BT-doped ceramics were far less investigated, and no critical size with enhanced properties has been reported to date.

With regard to BT-based solid solutions, BaZr_x_Ti_1−x_O_3_ (BZT) is very attractive for fundamental and applied research, due to its versatility of intrinsic properties by adjusting the zirconium/titanium ratio. The sharp ferroelectric to paraelectric phase transition for x < 0.15 becomes broader when increasing the substitution rate from 0.15 up to 0.25. An increase up to 0.40 leads to a relaxor behavior characterized by a frequency dependence of permittivity maximum. Moreover, similar behavior with a crossover to relaxor state was observed for composition at diffuse phase transition (x = 0.10) when reducing GS [19]. According to all of these previous results, such a material, with a ferroelectric-relaxor crossover tailored by both composition and grain size, presents a high potential in terms of application. It can be used as MLCCs, lead-free piezoelectric transducers, chemical sensors, microwave devices, etc. [17]. Until now, few papers have reported size-dependent properties of BZT ceramics, and all of them presented data for compositions with high Zr content (x > 0.10) [17,20]. The reported data have shown that the large grains undergo structural phase transition, while fine grains show diffuse phase transition. However, these data are for only a few grain sizes, and a systematic study of size effect on dielectric properties of BZT ceramics with GSs from nanometers to micrometers, prepared by the same processing technique, has never been performed.

Among BZT compositions, the x = 0.05 has the advantage of ferroelectric state at room temperature in the orthorhombic phase [21], which allows the study of the size dependence of dielectric properties in two ferroelectric phases: orthorhombic and tetragonal. This composition is also close to pure BT, and if a critical size in BT-based solid solutions exists, it is more likely to have a similar critical size in slightly doped BT.

Therefore, for the present paper, BaZr_0.05_Ti_0.95_O_3_ (BZT) ceramics with different grain sizes were prepared by solid-state synthesis in conjunction with conventional sintering, and the dielectric properties are systematically discussed. Furthermore, a correction method based on the finite element method was proposed to remove the effect of porosity in fine ceramics.

## 2. Materials and Methods

### 2.1. Sample Processing and Preparation

BZT nanopowders with an average particle size of 300 nm were prepared by the solid -state reaction using, as starting raw materials, high-purity nanopowders BaCO_3_ (Solvay Bario e Derivati, 99.9% purity, Massa, Italy), TiO_2_ (Evonik Degussa-P25, 99.9% purity, Frankfurt, Germany) and ZrO_2_ (Toho, 99.9% purity, Tokyo, Japan). The mixed powders were calcinated at 1000 °C for 4 h in order to promote the solid-state reaction. The calcinated powders were sieved and then manually re-milled. The resulting powders were compacted in cylinders (lengths of ~2–3 cm, diameters of ~1 cm) through cold isostatic pressing at 1500 bar, and the pellets were sintered at different temperatures (1150–1500 °C) and time intervals (2–24 h) in order to obtain different grain sizes. After sintering, the density was measured using Archimedes’ method.

### 2.2. Experimental Details

The microstructure of the ceramic samples was investigated using a high-resolution scanning electron microscope JSM 6390—Analytical Scanning Electron Microscope (JEOL Ltd., Tokyo, Japan). The average grain size was calculated considering size measurements on ~400 grains and using ImageJ Image Processing and Analysis Software (ImageJ: ver.1.52a, Wayne Rasband). The electrical measurements were performed using a parallel-plate capacitor configuration, by applying Ag electrodes onto the polished surfaces of disk-shaped ceramic samples. The dielectric measurements were carried out on an Agilent E4980A Precision LCR Meter (Santa Clara, CA, USA), for temperatures between 25 °C and 150 °C, in the frequency range of 20 Hz to 2 MHz, cooling at a rate of 5 °C/min from 150 °C down to room temperature.

## 3. Results and Discussion

A series of BZT ceramics with GS ranging from 0.45 µm to 135 µm and density ranging from 72% to 97% were prepared by conventional sintering from solid-state powder. The XRD patterns (not presented here) showed a single-phase composition with complete incorporation of zirconium in the perovskite lattice. The relative density was calculated according to the theoretical density of 6.0305 g/cm^3^ for BZT. The relative density and average GS of various BZT ceramics, sintered under different conditions, are presented in Appendix A (Appendix A. The ceramics with GS larger than 1.4 µm had a density between 94 and 97%, while the density of the ceramics with GS smaller than 1.2 µm decreased to 72%. While in the case of ceramics with density higher than 94% the effect of pores on dielectric properties was negligible, for the finest ceramics, where the porosity was larger, a model to correct the effect of porosity on dielectric permittivity was imperative.

The scanning electron microscopy (SEM) images of fracture surfaces presented in Figure 1a–f reveal the microstructural features of some selected BZT ceramics.

It is known that good densification of ceramics with small GSs, through classical sintering methods, is difficult to achieve [22]; only combined sintering methods such as spark plasma sintering and two-step sintering [6,23] have demonstrated highly densified nanoceramics. However, in the first approach, conventional sintering method allowed us to prepare a very large range of GSs, with very good densification for GS > 1.4 µm. The insets in the bottom-right corner of the SEM images (Figure 1a–f) show the histograms corresponding to the GS distribution of the selected BZT ceramics and the average GS. Ceramics with GSs between 1.6 and 50 µm present a non-homogeneous microstructure, having regions with grains around 1.5 µm and large grains around 35 µm that represent approximately 70% and 30% of the ceramic volume, respectively. To calculate the GS of these ceramics, we considered the weighted average of these two regions. More homogeneous microstructures were observed for ceramics with GS < 1.6 µm and GS > 50 µm.

The shapes of the pores presented peculiar modifications when reducing grain size. For ceramics with high GS (>1.44 µm) and small porosity levels (<10%), the pores were well isolated and intergranular, and their size was much smaller than the GS. For fine ceramics (Figure 1a), the pores were percolated and their average size (>1 µm) was much larger than the GS. For intermediate values of GS (e.g., in Figure 1b), the pores still presented a certain level of percolation, but their average size was comparable with the GS.

The evolution in temperature of real part of permittivity and dielectric loss for all investigated BZT ceramics is presented in Figure 2 for f = 10 kHz.

The data qualitatively show the tendency of the system towards a more diffuse phase transition when the average GS reduces to 0.45 µm. This behavior can be explained considering two causes: small grain size that induces a diffuse phase transition in BT-based ceramics [24] and porosity. As already shown in our previous papers, the porosity induces a diffuse ferroelectric phase transition in Ba-based ceramics [25]. All the samples present two transitions—one at around 45 °C (orthorhombic-tetragonal) and one at around 105 °C. Despite significant broadening of the permittivity peak, the position of permittivity maximum can still be considered as an indication of the ferroelectric–paraelectric transition temperature (Curie temperature, T_C_), even for the finest and most porous ceramics. As can be observed in Figure 2a, this temperature was not affected by GS. In the inset of Figure 2a, the evolution of the real part of permittivity with GS at two evocative temperatures is represented: at room temperature (25 °C), where this composition is in the orthorhombic phase, and at 70 °C, where it is in the tetragonal phase. For both temperatures, the permittivity seemed to exhibit a maximum for GSs between 1.5 µm and 10 µm. It is generally accepted that different mechanisms affect dielectric properties of pure and doped BT ceramics, including intrinsic size effect, grain boundaries, domain walls mobility and density and domain patterns [5]. Some of them are predominant at nanoscale (intrinsic effects and grain boundaries), others at larger grain sizes. Although it is accepted that the grain boundaries in BT-based ceramics have non-ferroelectric behavior and smaller permittivity, the dilution effect is observed especially in ceramics with grain sizes smaller than 0.3–0.5 µm [5]. In BT ceramics with grain sizes larger than 0.5 µm (as in our case), the stripe domain patterns with 90° domain walls are predominant [5]. Due to the fact that our ceramics are very close in composition to pure BT, we expect a similar contribution: (i) a combined contribution of grain boundary and domain patterns for ceramics with fine GSs and (ii) mainly domain configuration contribution for ceramics with GSs > 1.2 µm. The dielectric losses were below 5% in the investigated temperature range (Figure 2b). In particular, the tangent loss was significantly suppressed at temperatures above T_C_. The maximum of losses was obtained for ceramics with larger grains (> 50 µm) and higher density (~97%).

The main shortcoming of our ceramics was that, for the smaller GSs, a 20–30% porosity was present, and measured permittivity is strongly affected by this porosity, as shown in our previous papers [25,26].

In order to eliminate the role of porosity and to estimate the grain size-dependent permittivity values of the BZT ceramics (as it would be obtained in the case of fully dense ceramics), a finite element method (FEM) was employed. In former studies, we demonstrated that FEM simulations, even at a 2D level, are a much better alternative to effective medium approximations (Maxwell-Garnett or Bruggeman) to describe the effective permittivity in ferroelectric-based composites [27]. The first step in simulations was to generate appropriate 3D porous ceramics with similar microstructural features as revealed by SEM analysis (Figure 1). The virtual ceramic systems were created by generating spherical grains with a desired diameter (grain size) in random positions, until the desired density was reached. Additionally, the average sizes of the pores were imposed to be similar as revealed by SEM. This way, porous ceramics with grain sizes of 0.45 µm to 150 µm and porosity levels of 1% to 29% were generated. For example, the virtual 3D microstructures that correspond to the SEM images represented in Figure 1 (GSs of 0.45 µm, 1.07 µm and 1.44 µm) are represented in Figure 3a–c.

For low porosity levels (<6%) that correspond to relatively large grain sizes (>1.44 µm), the size of the pores was generally considered smaller than the grain size and located at the boundary between several grains, like in the example of the microstructure represented in Figure 3c. As shown by SEM images (Figure 1), in the case of ceramics with lower GSs (<1.32 µm), the size of the pores was comparable or even higher than the GS, and this aspect was taken into account when generating the corresponding 3D ceramic systems (Figure 3a,b, respectively).

The next step in the FEM simulations was to solve the Laplace equation: $div\cdot \left(\varepsilon \;grad\varphi \right)=0$, where φ is the local potential and ε is the relative permittivity of the pores (equal to 1) or of the bulk ceramic (*ε_BZT_*). *ε_BZT_* is the permittivity of the bulk BZT as it would be obtained in the case of dense ceramics (no porosity). As in dense ceramics, *ε_BZT_* incorporates other important effects related to grain boundaries, domain structures, polar nanoregions, etc., which are not explicitly introduced in the present 3D simulations. The boundary conditions in the FEM simulations were considered to be the conditions specific to a parallel plate capacitor subjected to a voltage between two electrodes (represented in red and blue in Figure 3). After calculations of the local electric field distributions (Figure 3d–f) for certain values of *ε_BZT_*, the effective permittivity of the system was determined from the total energy of the capacitor according to a procedure presented in [26,27]. The aim of the present simulations was to find the permittivity of the bulk ceramic (*ε_BZT_*) for each system that corresponds to the experimentally measured effective permittivity value (Figure 2). The values of *ε_BZT_* were determined by performing simulations for different *ε_BZT_* values in the range of 100–10,000, following a bisection method of the searching interval until the desired effective permittivity was reached. The local field images represented in Figure 3d–f correspond to the final iteration steps in the bisection method and to the effective permittivity values determined by experimental measurements at room temperature (Figure 2). The simulations presented in Figure 3 demonstrated that the porous BZT ceramics are characterized by complex configurations of the local electric fields, which are mostly affected by two factors: (i) the type of the pores (percolated for low grain sizes and isolated for high grain sizes) and (ii) the porosity level (as a dilution effect).

A real ferroelectric ceramic is more complex than the di-phase composite model proposed in this work. A more realistic approach should model a porous ferroelectric ceramic as a tri-phase composite with grain bulk, grain boundaries and pores, but this is very difficult to be realized by 3D FEM simulations, even if the number of elements is very high. For example, in the simulations presented in Figure 3, the number of the elements is 24 million, but the average size of the elements is still very high (over 60 nm), and the meshing grid cannot properly describe the boundaries. However, using a Curie–Weiss calculation of the dielectric permittivity at 200 °C (Appendix A and Appendix A
Appendix A), we estimated the influence of the grain boundaries at less than 20% in the finest samples. A more facile 2D simulation approach (Appendix A) demonstrated that, at such dilution effect of the grain boundaries, a di-phase composite describes well, at least qualitatively, the local electric field on bulk and pores. Several comparisons between a di-phase approach and a tri-phase approach (Appendix A
Appendix A), used to simulate the effective permittivity of the same 2D porous ceramic structures, demonstrated the accuracy and the correctness of the di-phase model in computing the effective permittivity for all GSs. In addition, the 2D simulations presented in Appendix A (Appendix A) demonstrated that the local electric fields in pores are described by the di-phase and tri-phase approaches in a similar way, irrespective of GS.

Despite the simplifications, using 3D FEM simulations we were able to eliminate the role of porosity and estimate the permittivity dependence of the ceramic bulk on grain size. Two such dependences were determined at temperatures of 25 °C and 70 °C; these are represented in Figure 4 and compared with the experimental results.

The major improvement of the *ε*(*GS*) dependences was noticed especially at GSs smaller than 1.3 µm, where the measured values of permittivity were abnormally much smaller than the permittivity values. In addition, the experimentally determined *ε*(*GS*) dependences presented an abnormal minimum at GS = 1.3 µm, which was eliminated after performing the FEM corrections. The FEM simulations also demonstrated, at a quantitative level, that the decreases of permittivity at GSs < 1.4 µm, experimentally observed, is not just a porosity effect, and that the maximum of permittivity at GSs between 1.5 µm and 10 µm is a grain size effect.

## 4. Conclusions

In this work, a systematic investigation of the grain size effects on the dielectric properties of BZT dense ceramics with x = 0.05 and grain sizes between 0.45 and 135 µm is presented. The ceramics were prepared from solid-state powders sintered at different times and temperatures for obtaining grain sizes between 0.45 and 135 µm. The effects of grain size on dielectric properties were systematically explored, and it was found that dielectric permittivity was significantly influenced by grain size, reaching a maximum value around 1.5–10 µm and then rapidly dropping with a further increase in grain size. A FEM method was employed to eliminate the role of porosity and to estimate the grain size-dependent permittivity values of the BZT ceramics (as it would be obtained in the case of fully dense ceramics). The results suggest that it is possible to have a critical size in slightly doped BT ceramics with enhanced functional properties for GSs between 1.5 and 10 µm.

## Figures and Tables

**Figure 1 materials-13-04386-f001:**
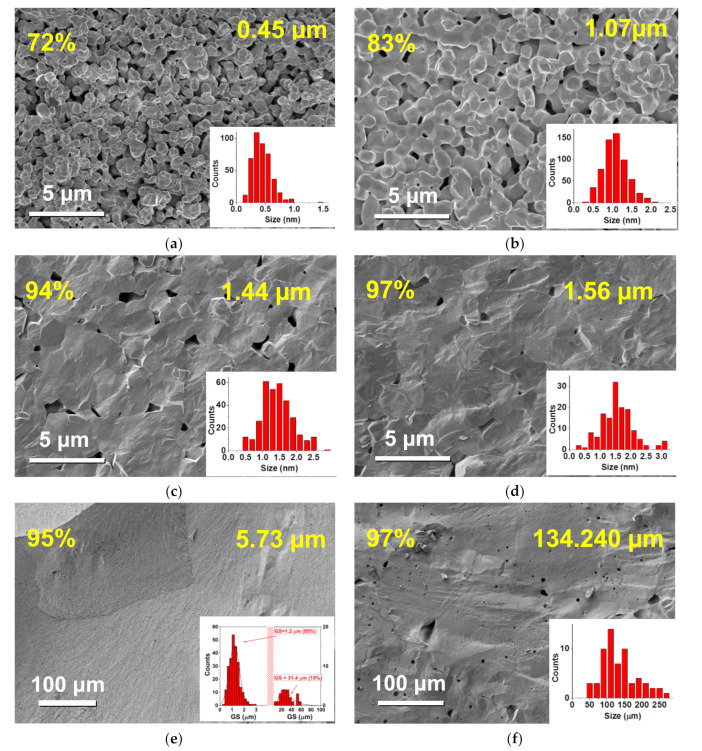
The effect of processing conditions on microstructure: (**a**–**f**) micrographs and histograms of BaZr_0.05_Ti_0.95_O_3_ (BZT) ceramics.

**Figure 2 materials-13-04386-f002:**
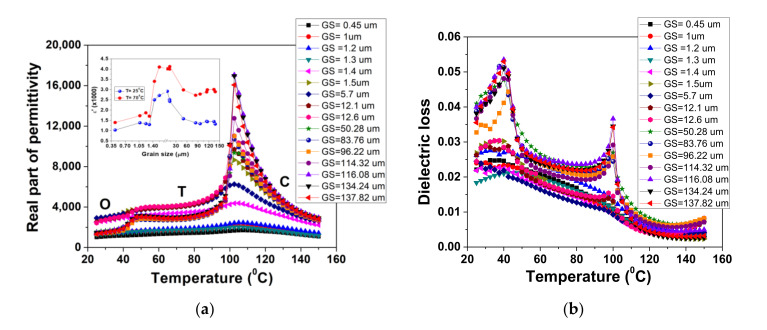
Temperature dependence of (**a**) real part of permittivity and (**b**) dielectric loss in BZT ceramics with different grain sizes at f = 10 kHz. Inset in (**a**): grain size dependence of dielectric permittivity at two selected temperatures.

**Figure 3 materials-13-04386-f003:**
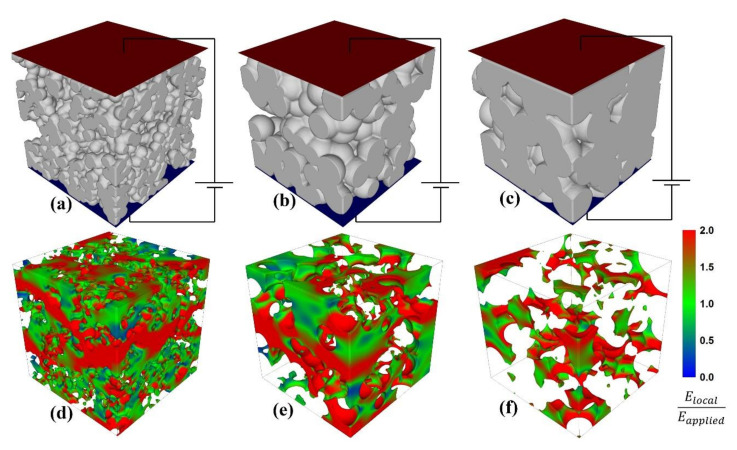
Representations of local field images in pores: (**d**–**f**) in color scale corresponding to 3 ceramics with different grain sizes and porosity levels; (**a**–**c**) in a parallel-plate capacitor configuration (red surface—top electrode; blue surface—bottom electrode). The grain sizes and porosity levels of the systems are 0.45 µm and 28% (**a**,**d**), 1.07 µm and 17% (**b**,**e**) and 1.44 µm and 6% (**c**,**f**).

**Figure 4 materials-13-04386-f004:**
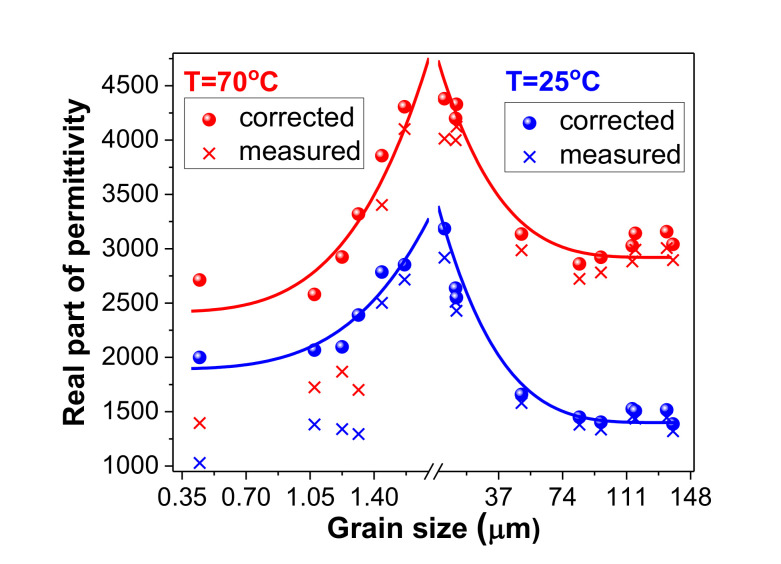
Estimated dependences of the dielectric constant on grain sizes by finite element method (FEM) simulations for fully dense BaZr_0.05_Ti_0.95_O_3_ ceramics (the continuous lines are guides for the eye) and comparisons with experimental data.

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
