# Peer review of "Scale-Dependent Dielectric Properties in BaZr0.05Ti0.95O3 Ceramics"

_materials, 2020, doi:10.3390/ma13194386_

Round 1

Reviewer 1 Report

If authors have some data concerning porosimetry of the studied samples it would nice to see them in Supplementary Materials.

The results of the provided 3D FEM simulations should be in consistence with such data.

Some "polishing" work is recommended for English. For example, in the line 26 the word "need" is mentioned twice.

Author Response

Dear reviewer,

Please let us to express our gratitude for the opportunity to consider our manuscript for publication.  Please find in the following the list of all answers. 

-If authors have some data concerning porosimetry of the studied samples it would nice to see them in Supplementary Materials.

            Unfortunately, the only information we have about porosity are derived from the measurements of relative density (Archimedes method) and from Scanning Electron Microscopy (SEM). The samples were investigated in fracture and we were not able to determine other parameters of the pores like average size. However, we introduced new sentences before Fig. 1 in which the pores and the microstructural characteristics, derived from SEM investigations, are much better described.

-”The results of the provided 3D FEM simulations should be in consistence with such data.”

            In order to better explain how we took into consideration the role of porosity (including the pores shapes and microstructural characteristics revealed by SEM for different grain sizes), we performed some modifications in the manuscript (both in the experimental and modeling parts)

- ”Some "polishing" work is recommended for English. For example, in the line 26 the word "need" is mentioned twice.”

            The manuscript was checked again and we hope the English was improved. The comments of yours and of the other reviewers were very helpful.

Reviewer 2 Report

The content of the manuscript falls within the topics covered of Materials Journal. The paper is well presented, and the results are discussed with profound physical concepts; the experimental data were fitted using theoretical models. The Paper can be accepted with without revision

Author Response

Dear reviewer,

The manuscript has undergone improvements and the English language was verified.

Reviewer 3 Report

The manuscript carefully and systematically presented the particle size effects of BZT. Porosity consideration is essential and excellent part of this work. Therefore,  please present Fig. 4 and the inset of Fig. 2(a) to be compared clearly. Fig. 2(a) data may be underlaid. Please plot the porosities and grain sizes in Table S1 together so that any correlation between grain size and/or porosity and permittivity can be found.

The porosity effects have been taken into account using Bruggeman equation. The estimate may be compared and discussed.

Other minor points are listed as below. 

... was assigned(-> attributed or ascribed) to internal..
If (-> While? Whereas?) in the case of pure BT ceramics, the size...

Last year(!), Dupuy... : (Very) Recently

he effect of grain size and processing technique (phase homogeneity)... : The parenthesized part does not fit!

Please explain how the system considered can be used as "chemical sensors"

... are systematically discussed. Meanwhile(?), a correction method ...

... no pores were positioned at the junctions between grains, like in the example of the microstructure represented in Figure 3(c). : The meaning is not clear. The pores are shown to locate at the triple junctions in Fig. 3(c). Is "junction" meant for the grain boundaries?

The local field images represented in Figures 3(c,d,e)-> d,e,f

... more realistic approach should model a porous ferroelectric ceramic as a tri-phase composite with grain bulk, grain boundaries and pores ... : What thickness (or volume) of the grain boundaries is assumed? Is it relevant? That is, the thickness is not needed? So the grain boundary effects are the discontinuity in the field lines?

Author Response

Dear reviewer,

 Please let us to express our gratitude for the opportunity to consider our manuscript for publication. Please find in the following the list of all changes that were made to comments.

- The manuscript carefully and systematically presented the particle size effects of BZT. Porosity consideration is essential and excellent part of this work. Therefore,  please present Fig. 4 and the inset of Fig. 2(a) to be compared clearly. Fig. 2(a) data may be underlaid. Please plot the porosities and grain sizes in Table S1 together so that any correlation between grain size and/or porosity and permittivity can be found.

            The experimental data were also included in Fig. 4, for comparisons. A graph showing the dependence of the porosity on GS was added in the Supplementary material (Fig. S1).

- The porosity effects have been taken into account using Bruggeman equation. The estimate may be compared and discussed.

            We have checked before not only Bruggeman equation, but also other Effective Medium Approximations like Maxwell-Garnett or Lichtenecker and we observed that they provide different results than the FEM simulations. None of these equations is able to describe the particularities of a porous ceramic. Maxwell-Garnett equation is correct for particulate composites (with spherical isolated inclusions in matrix), while Bruggeman or Lichtenecker equations can be applied for composites with randomly mixed phases. A porous ceramic is not a random mix of bulk and pore air, but it is characterized by connected (partially sintered) grains. Bruggemen equation strongly underestimates the effective permittivity of porous ceramics, by comparison with our approach. We added in the manuscript a citation to a former work (Ciomaga, C.E.; Olariu, C.S.; Padurariu, L.; Sandu, A.V.; Galassi, C.; Mitoseriu, L. Low field permittivity of ferroelectric-ferrite ceramic composites: Experiment and modeling. J. Appl. Phys. 2012, 112, 094103) which includes a discussion about the differences between these approaches and shows that FEM simulations is the most accurate method for ferroelectric-based ceramics.

- Other minor points are listed as below.

... was assigned(-> attributed or ascribed) to internal.

We modified in text with “attributed”.

If (-> While? Whereas?) in the case of pure BT ceramics, the size...

We replaced “if” with “while”.

Last year(!), Dupuy... : (Very) Recently

We changed with “recently”.

The effect of grain size and processing technique (phase homogeneity)... : The parenthesized part does not fit!

The sentence was wrong and we corrected the mistake. The new sentence is: "Recently, Dupuy et al. reported the effect of grain size and phase homogeneity on the ferroelectric properties of a more complex solid solution of BT - (Ba,Ca)(Ti,Zr)O3".

Please explain how the system considered can be used as "chemical sensors"

A chemical sensor contains two functional units: a receptor and a transducer part.  These materials (BaTiO3- based solid solutions) can be used as transducer parts. These materials change their electrical properties (modified permittivity, change in conductivity due to reversible redox processes, etc.) by interacting with the analyte.

... are systematically discussed. Meanwhile(?), a correction method ...

We change in text with “Furthermore”.

... no pores were positioned at the junctions between grains, like in the example of the microstructure represented in Figure 3(c). : The meaning is not clear. The pores are shown to locate at the triple junctions in Fig. 3(c). Is "junction" meant for the grain boundaries?

The sentence was wrong and we corrected the mistake. Indeed, the pores are located at boundaries between grains for ceramics with large GS.

The local field images represented in Figures 3(c,d,e)-> d,e,f

We corrected with Fig. 3 (d,e,f).

... more realistic approach should model a porous ferroelectric ceramic as a tri-phase composite with grain bulk, grain boundaries and pores ... : What thickness (or volume) of the grain boundaries is assumed? Is it relevant? That is, the thickness is not needed? So the grain boundary effects are the discontinuity in the field lines?

The thickness (volume) of the grain boundaries is not important to be explicitly defined for the scope of our simulations which is to find the permittivity of fully dense ceramics (bulk+grain boundaries, but without pores). This paragraph was introduced in manuscript to make known that grain boundary dilutions effects were not ignored in our study. The effective permittivities we proposed to determined by 3D FEM simulation (the di-phase approach) include this effect, irrespective of the grain boundaries volume. The correctness of this simplification (not considering the boundaries in the 3D system as a third separated phase) is discussed in the Supplementary material for different amounts of boundaries.